# Predictive Factors for 24-h Survival After Perioperative Cardiopulmonary Resuscitation: Single-Center Retrospective Cohort Study

**DOI:** 10.3390/jcm14020599

**Published:** 2025-01-17

**Authors:** Soontarin Chungsaengsatitayaporn, Tanyong Pipanmekaporn, Jiraporn Khorana, Prangmalee Leurcharusmee, Settapong Boonsri, Visith Siriphuwanun

**Affiliations:** 1Department of Anesthesiology, Faculty of Medicine, Chiang Mai University, Intavarorote Rd., Muang Chiang Mai District, Chiang Mai 50200, Thailand; khunyimkha@gmail.com (S.C.); prangmalee.l@cmu.ac.th (P.L.); settapong.b@cmu.ac.th (S.B.); visith_cmu@yahoo.com (V.S.); 2Department of Biomedical Informatics and Clinical Epidemiology, Faculty of Medicine, Chiang Mai University, Chiang Mai 50200, Thailand; jiraporn.kho@cmu.ac.th; 3Division of Pediatric Surgery, Department of Surgery, Faculty of Medicine, Chiang Mai University, Chiang Mai 50200, Thailand; 4Clinical Surgical Research Center, Department of Surgery, Faculty of Medicine, Chiang Mai University, Chiang Mai 50200, Thailand

**Keywords:** cardiopulmonary resuscitation, perioperative, cardiac arrest, prognosis, retrospective studies, survival

## Abstract

**Background:** Perioperative cardiac arrest (POCA) remains a major challenge in surgical settings, with low survival after cardiopulmonary resuscitation (CPR). This study aims to identify predictive factors for 24 h survival after CPR and cause of POCA. **Method:** A retrospective, single-center study was conducted on patients aged ≥18 years who experienced POCA and received CPR in the operating room or within 2 h postoperatively at Chiang Mai University Hospital from 2010 to 2019. The multivariable analysis of independent survival predictors was performed using risk regression models. **Results:** There were 288 cases of cardiopulmonary arrest requiring CPR, with 61 patients surviving. Significant predictors of survival after perioperative CPR included the American Society of Anesthesiologists physical status classification (ASA) 1–2 (RR 2.53; 95%CI 1.69–3.77; *p* < 0.001), preoperative hemoglobin ≥ 8 g/dL (RR 2.27; 95%CI 1.05–4.89; *p* = 0.036), preoperative oxygen saturation ≥ 90% (spontaneous breathing in room air) (RR 3.19; 95%CI 1.21–8.41; *p* = 0.019), initial end-tidal carbon dioxide between 35 and 45 mmHg (RR 1.55; 95%CI 0.98–2.44; *p* = 0.059), and duration of CPR ≤ 30 min (RR 3.68; 95%CI 1.51–8.98; *p* = 0.004). The major cause of POCA was hypovolemia (74.30%). **Conclusions:** This study identifies several critical predictors for 24 h survival following POCA, which can inform pre-operative optimization and perioperative management. Timely interventions, such as blood transfusions and volume resuscitation, are crucial in improving survival outcomes, particularly in trauma and high-risk patients. Further multi-center studies are needed to validate these findings and explore long-term outcomes to refine perioperative cardiac arrest management.

## 1. Introduction

Perioperative cardiopulmonary arrest (POCA) is a life-threatening event that occurs in the operating room or shortly after surgery [1,2] with outcomes that often include survival with or without complication or death [3,4,5,6,7,8]. POCA is distinct from other cardiac arrests in the hospital setting arrests due to its close association with anesthesia and surgical interventions, which are challenges for timely diagnosis, intervention, and management. Cardiac arrest refers to the absence of circulation, requiring cardiopulmonary resuscitation (CPR) whether by chest compression or, in rare cases, open cardiac massage [6,8,9,10,11]. Although POCA is relatively rare compared to other adverse events, its incidence and outcomes vary globally, often reflecting the complexity of healthcare environments [12]. Developed countries report better post-CPR outcomes due to advanced healthcare systems, resources, and comprehensive research support [13], while the incidence of POCA in these settings is reported to be between 2.63 and 6.8 per 10,000 cases [9,14,15,16,17]. In Thailand, the Perioperative Anesthetic Adverse Events in Thailand (PAAd Thai) study reported a higher incidence of POCA at 15.5 per 10,000 cases, signaling a need to understand the risk factors and outcomes more deeply [3].

The causes of POCA are often sudden and directly linked to intraoperative events, including hypovolemia from surgical blood loss, airway obstruction, hypoxia, and anesthetic-related cardiovascular suppression. These intraoperative events demand rapid, specialized interventions to prevent irreversible damage or death, setting POCA apart from general in-hospital cardiac arrests, which often result from chronic conditions and present in less-controlled environments like general wards. Continuous monitoring such as continuous end-tidal CO_2_ (ETCO_2_), pulse oximetry, and electrocardiography (ECG), and the readiness and awareness of anesthesiologists, surgeons, and trained nursing staff in resuscitation enables immediate resuscitation efforts [18,19,20] which lead to a quicker response, potentially improving the chances of successful intervention by utilizing the readily available medical resources in operating rooms [3,12,21,22,23]. Although the controlled environment and rapid response capabilities provide some advantage for POCA cases, survival rates are far from consistent, with outcomes influenced by a complex interplay of patient-specific factors, such as pre-existing comorbidities and the urgency of the surgery [8,14,18,19,20]. Despite the available resources, research on POCA survival predictors remains inconclusive, with findings on factors like The American Society of Anesthesiologists (ASA) physical status, surgical urgency, and initial arrest rhythms often varying widely due to the differences in the study methodologies and patient characteristics [11,22,24,25,26,27,28]. Furthermore, while short-term survival (within 24 h post-CPR) is an essential indicator of immediate intervention success, few studies have focused on this critical timeframe [11,29,30]. For perioperative clinicians, understanding specific predictors within the first 24 h is crucial for preoperative optimization, real-time intraoperative monitoring, and tailoring resuscitation efforts to patient needs, thus highlighting an important gap in current knowledge.

This study addresses these gaps by analyzing the key clinical predictors associated with 24 h survival following POCA. By examining detailed variables such as ASA classification, preoperative oxygen saturation, initial ETCO_2_ levels, and CPR duration, this study aims to provide actionable insights that can refine perioperative protocols and resuscitation practices. Ultimately, identifying these predictors will support better risk stratification, improve clinical decision making, and contribute to a safer perioperative environment, enhancing patient outcomes after cardiac arrest.

## 2. Material and Methods

### 2.1. Study Design

This retrospective cohort study was conducted according to the STROBE guidelines, ensuring the transparent reporting of observational studies. The Ethics Committee of the Faculty of Medicine at Chiang Mai University approved this study (Research ID: 8823, Study Code: ANE:2565-08823), and a waiver of informed consent was granted due to the retrospective design.

### 2.2. Setting

Data were collected from Chiang Mai University Hospital, Thailand, covering POCA cases between January 2010 and December 2019. A summary of the study protocol and collected data is available in Appendix A.

### 2.3. Participants

This study included adult patients aged 18 years and older who experienced POCA during non-cardiac surgeries under general anesthesia, regional anesthesia, or monitored anesthesia care (MAC). Exclusion criteria encompassed patients undergoing organ transplantation, pregnant patients, and cases with incomplete POCA documentation.

### 2.4. Variables

Data extraction and analysis were carried out by the research team, consisting of two research assistants and anesthesiologists, who reviewed each case to ensure consistency and accuracy. Baseline characteristics were carefully extracted from anesthesia records, including demographic data (age, gender), The American Society of Anesthesiologists physical status, and documented preoperative comorbidities. Comorbidities noted included respiratory diseases, diabetes, hypertension, cardiovascular disease, and renal dysfunction. Additionally, airway management strategies, recorded prior to entering the operating room, were documented as part of the baseline data.

Surgical data encompassed the nature of the surgery (emergency vs. elective), specific surgical sites (e.g., intrathoracic, intra-abdominal, major vascular, intracranial, and major orthopedic procedures), trauma status, specifying whether cases involved multiple injuries, whether perioperative data were collected on blood loss (measured in milliliters), the need for blood transfusion, and patient positioning during surgery. Preoperative laboratory values, obtained from patient records, included blood glucose, hemoglobin, platelet count, the neutrophil–to-lymphocyte ratio (NLR), serum albumin, creatinine, sodium, potassium, and 12-lead electrocardiogram (ECG) readings to provide a comprehensive baseline profile.

Intraoperative parameters and vital signs: Upon arrival in the operating room (OR), initial vital signs were recorded to establish baseline conditions before any cardiac events. These parameters included heart rate (HR), systolic blood pressure (SBP), diastolic blood pressure (DBP), mean arterial pressure (MAP), and oxygen saturation (SpO_2_) with levels over 90% considered stable, for patients breathing spontaneously in a room air environment. ETCO_2_ was measured immediately following successful intubation. This set of baseline vital signs was crucial in assessing the patient condition upon entry into the OR.

Details of POCA events were characterized by detailed timing, diagnosis, and physiological status at the moment of arrest. The occurrence of POCA was noted with respect to working hours (08:00 a.m.–04:00 p.m.) or non-working hours (04:00 p.m.–08:00 a.m.) to account for potential variations in resource availability. Diagnosis included initial cardiac rhythms, categorized as shockable (ventricular fibrillation, pulseless ventricular tachycardia), or non-shockable (asystole, pulseless electrical activity). The duration of CPR was measured from the onset of absent cardiac rhythm until the return of spontaneous circulation (ROSC), providing insight into the effectiveness of resuscitative efforts.

Causes and contributing factors of POCA: The authors used the mnemonic “5Hs and 5Ts”, which serves to identify reversible factors in cardiac arrest as shown in Figure 1 [31].

Other notable causes outside the “5Hs and 5Ts” framework were also recorded, such as local anesthetic systemic toxicity and malignant hyperthermia, a rare but critical anesthetic-related syndrome that is a significant cause of anesthetic-related morbidity and mortality in an otherwise healthy patient [24,32,33].

### 2.5. Data Sources/Measurement

Data were extracted from electronic and paper medical records, including anesthesia records, CPR records, and laboratory reports, with measurements standardized according to previous studies (Appendix A) [34,35,36,37,38,39,40,41]. Initial vital signs and ETCO_2_ were recorded upon arrival in the operating room, ensuring that baseline values reflected pre-arrest conditions as closely as possible.

### 2.6. Bias

Selection bias was minimized by defining clear inclusion and exclusion criteria and utilizing standardized data collection protocols. The research assistants reviewed and validated data entries to maintain consistency and reduce potential errors in recorded measurements.

### 2.7. Study Size

The sample size was estimated using STATA version 16.0 (Stata Corp LP, College Station, TX, USA). A two-proportion test with an alpha level of 0.05 and a power of 0.80 indicated a requirement of 77 participants per group, resulting in a minimum total of 154 patients. This study included 288 eligible cases, ensuring robust statistical power to detect differences across predictor variables.

### 2.8. Statistical Methods

Data analysis was conducted using STATA version 16.0 (Stata Corp LP, College Station, TX, USA). Categorical data are presented as frequencies and percentages, while continuous data are reported as mean with standard deviation or median with interquartile range, depending on the distribution. The Shapiro–Wilk test was applied to assess the normality of continuous data. Categorical variables were compared using Fisher’s exact test, and continuous variables were analyzed using the Student’s *t*-test or the Mann–Whitney U test, based on data distribution.

Variables with *p*-values less than 0.1 in univariable analysis were selected for further examination using stepwise multivariable risk regression to determine risk ratios (RRs), with additional variables identified from relevant literature. Statistical significance was defined as a *p*-value of less than 0.05.

Subgroup analysis was performed to compare the survival rates among patients aged 65 years or older versus those younger than 65, as age was associated with higher survival rates. Specifically, for patients undergoing trauma surgery, detailed age comparisons are presented in Appendix A, subgroup analysis between age and type of surgery is presented in Appendix A, and subgroup analysis between age and the type of operations in the trauma group (149 patients) is presented in Appendix A.

## 3. Results

In this 10-year retrospective study, a total of 181,905 patients underwent surgery with anesthesia, including general anesthesia, regional anesthesia, and MAC. The incidence of POCA was found to be 19.95 per 10,000 cases (363 of 181,905 patients). After applying the inclusion criteria for non-cardiac surgeries with POCA, 363 cases were identified, among which 75 cases met the exclusion criteria, leaving 288 patients included in the final analysis. Among these, 61 patients (21.18%) survived within 24 h following POCA (Figure 2).

All patients who experienced POCA had received general anesthesia, with only one initially receiving an infraclavicular brachial plexus block and two others undergoing MAC. Patients aged 65 years and older demonstrated a higher survival rate compared to those younger than 65 years (RR 1.67; 95%CI 1.07–2.59; *p* = 0.023). Patients with ASA physical status classification 1–2 had a higher likelihood of survival compared to those with ASA 3–5 (RR 4.92; 95%CI 3.46–6.99; *p* < 0.001). Patients undergoing non-trauma surgery demonstrated a higher survival rate compared to surgery for trauma with multiple injuries (RR 2.04; 95%CI 1.26–3.28; *p* = 0.003). Patients with no preoperative signs of shock before experiencing POCA had better survival rates than those who were in shock upon arrival in the operating room (RR 2.41; 95% CI 1.34–4.33; *p* = 0.003) (Table 1 and Table 2).

Regarding the preoperative laboratory investigation, hemoglobin levels of ≥8 g/dL (RR 2.96; 95%CI 1.40–6.23; *p* = 0.04) was the most significant predictor linked to survival (Table 3).

The vital signs upon arrival to the operating room are crucial predictors of CPR outcomes, with preoperative SPO_2_ being particularly important. Patients with SPO_2_ ≥ 90% had a significantly higher survival rate compared to those with preoperative SPO_2_ measured by pulse oximetry while breathing room air less than 90% (RR 4.82; 95% CI 1.99–11.61; *p* < 0.001). Patients with initial ETCO_2_ levels between 35 and 45 mmHg had a better chance of survival compared to those with levels outside this range (RR 2.41; 95% CI 1.53–3.80; *p* < 0.001) (Table 4).

Patients with an initial shockable rhythm, such as pulseless ventricular tachycardia (pVT) or ventricular fibrillation (VF), had better survival outcomes compared to those with an unshockable rhythm (RR 1.65; 95% CI 0.99–2.73; *p* = 0.050). Additionally, a CPR duration of less than 30 min was significantly associated with improved survival compared to CPR lasting more than 30 min (RR 4.66; 95% CI 1.75–12.42; *p* = 0.002) (Table 5).

The leading cause of POCA was hypovolemic shock, as presented in Table 6.

The multivariable risk regression analysis identified strong predictors of survival within 24 h after receiving CPR (Figure 3). These predictors include age greater than 65 years (RR 1.73; 95%CI 1.11–2.09; *p* = 0.014), ASA 1–2 (RR 2.71; 95%CI 1.80–4.08; *p* < 0.001), preoperative SPO_2_ measured by pulse oximetry while breathing room air greater than 90% (RR 3.26; 95%CI 1.23–8.64; *p* = 0.018), initial ETCO_2_ levels within the normal range (RR 1.63; 95%CI 1.03–2.57; *p* = 0.035), and a CPR duration equal to or less than 30 min (RR 3.8; 95%CI 1.59–9.11; *p* = 0.003).

## 4. Discussion

This study found a 24 h survival rate of 21% following perioperative CPR (Figure 2), emphasizing the critical nature of the condition. This survival rate is significantly lower than previously reported rates, which range from 32% to 51.4% in the OR settings within 24 h after perioperative CPR [6,10,42,43]. Previous studies indicate that survival rates for cardiac arrest are significantly higher in the OR compared to general wards or ICUs. This is primarily due to the swift recognition of patient condition changes through continuous monitoring and the prompt initiation of resuscitation measures [12,44,45]. The prior research typically examines short- or long-term survival outcomes, but the identification of predictive factors for survival rarely remains [15,46,47]. The causes of POCA in the OR are distinct from those in other settings, frequently arising from anesthetic effects, patient comorbidities, and the complexity of surgical procedures [6,44]. Nonetheless, inadequate preoperative assessments remain a critical challenge, as they can contribute to suboptimal decisions and poor intraoperative outcomes, even in this resource-intensive environment [12,48]. Our findings aim to improve risk stratification and guide evidence-based decision-making during resuscitation efforts by examining factors such as physiological parameters and preoperative conditions. These insights not only provide a foundation for enhancing perioperative protocols but also underscore the importance of targeted interventions to optimize patient outcomes following cardiac arrest. Ultimately, these findings contribute to a more comprehensive understanding of survival determinants in POCA and highlight opportunities to improve perioperative care practices.

Our study demonstrates that patients aged 65 and older, particularly those undergoing elective or non-emergent surgeries, have significantly higher survival rates. This contrasts with much of the existing literature, where advanced age generally correlates with poorer survival outcomes following cardiac arrest [16,22,23,26,28]. In our cohort, older patients were frequently admitted for planned, elective procedures, allowing for better preoperative optimization and lower ASA scores. Elderly patients in this study demonstrated greater stability and higher survival rates compared to younger trauma patients, who faced a markedly high 24 h mortality rate (89.84%) due to severe, multi-system injuries requiring urgent intervention (see Appendix A).

The ASA physical status score is a preoperative rapid risk assessment tool routinely used for all anesthetized patients to predict perioperative complications and mortality [47]. The ASA score of less than 3 was associated with a 14% increase in survival probability [2], highlighting the role of lower ASA scores in improving outcomes. Conversely, higher ASA scores indicate compromised physiological reserves, often reflecting the severity of acute and high-risk conditions [10,18,22,23,26,30,45,49,50,51].

Non-trauma surgery has low risks associated with trauma surgeries, where controlled surgical environments and thorough preoperative planning contribute to significantly higher survival rates. Because of trauma surgeries often conducted under emergency conditions, limit the opportunity for stabilization, amplifying perioperative risks [16]. Consistent with prior studies [12,22,25,26], which showed trauma as the primary cause of POCA and the third leading contributor to mortality among younger patients, and that injuries frequently involved the brain, chest, abdomen, or major bones, often resulting in hypovolemic shock, cardiac tamponade, or respiratory failure [52,53].

No preoperative sign of shock upon arrival to the operating room emerged as a crucial predictor of survival following POCA. Patients presenting without clinical signs of shock demonstrated a significant survival advantage, aligning with studies emphasizing the importance of stable hemodynamics for effective responses to surgical stress and resuscitation efforts [19,23,54]. In the perioperative context, it is important to recognize that shock is characterized by severe hypotension, impaired perfusion, and organ dysfunction, reflecting that hemodynamic instability and borderline low BP without meeting the criteria for shock may still indicate underlying instability. Such cases, particularly in trauma patients, may represent a compensatory phase where perfusion is maintained through increased vascular resistance, masking the early stages of hemodynamic compromise [19,44].

Preoperative hemoglobin levels are critical for assessing patient stability before surgery. In this study, hemoglobin levels above 8 g/dL were strongly associated with improved 24 h survival rates because of adequate oxygen-carrying capacity in maintaining organ perfusion and supporting effective resuscitation during CPR [55,56,57]. However, it is important to note that preoperative hemoglobin levels may not fully reflect the patient’s status during POCA, particularly in trauma cases where massive blood loss can lead to a rapid decline in hemoglobin levels. This relationship between hemoglobin levels and survival highlights the necessity of preoperative optimization, especially for high-risk patients undergoing trauma surgeries where significant bleeding is anticipated. Maintaining sufficient preoperative hemoglobin reserves can improve physiological resilience to perioperative stress and enhance survival outcomes in crises [58].

Preoperative SPO_2_ levels above 90% with spontaneous breathing on room air upon arrival to the OR was identified as a significant predictor of 24 h survival following POCA. Higher SPO_2_ levels were associated with better hemodynamic stability under perioperative stress, reflecting adequate baseline oxygen reserves [59]. Conversely, low preoperative SPO_2_ (<90%), especially in trauma cases, was linked to compromised oxygen delivery to vital organs. This condition, compounded by intraoperative blood loss, exacerbates hypoxia, decreases perfusion, and heightens the risk of hypovolemia and myocardial ischemia due to reduced venous return, stroke volume, and cardiac output [60]. Preoperative SPO_2_ ≥ 90% serves as an actionable and easy-to-use parameter for identifying high-risk patients and optimizing perioperative care [25,26].

Initial ETCO_2_ values ranging from 35 to 45 mmHg, which were measured immediately following successful intubation, were strongly associated with survival, underscoring their role as a real-time marker of effective ventilation and perfusion. This parameter provides important feedback on the adequacy of CPR efforts, allowing clinicians to adjust resuscitative interventions to optimize outcomes and achieve ROSC [12,61]. The importance of ETCO_2_ monitoring in POCA situations lies in its dual function as a diagnostic and predictive tool, enabling the early recognition of ineffective resuscitation and guiding timely adjustments to improve outcomes. These findings are consistent with the broader utility of ETCO_2_ in emergency and perioperative settings, emphasizing its value in supporting high-quality resuscitation efforts. Combined with other predictors such as high SPO_2_ and stable hemoglobin levels, ETCO₂ forms part of an integrated set of perioperative indicators that can be continuously monitored and optimized [12]. Together, these parameters provide clinicians with actionable insights into the patient’s physiological status, enhancing their ability to deliver precise, effective care, and ultimately improving survival rates in POCA scenarios.

The presence of an initial shockable rhythm, such as pVT or VF, was significantly associated with improved survival outcomes, likely due to their distinct pathophysiology. Shockable rhythms reflect a state of electrical disorganization in the myocardium, which, if promptly treated with defibrillation [13,45,62,63], can be rapidly converted into an organized and perfusing rhythm. The rapid recognition of shockable rhythms enables targeted interventions, such as defibrillation, to provide circulation support and good outcomes [25,44,50].

The duration of CPR ≤ 30 min was associated with improved outcomes. This finding is consistent with prior studies demonstrating that shorter resuscitation times contribute to better neurological and physiological recovery, likely due to the earlier restoration of ROSC and reduced exposure to hypoxic injury [29,63,64]. The OR environment is particularly suited for continuous physiological monitoring, including blood pressure, EKG, SpO_2_, ETCO_2_, and in cases of regional anesthesia or MAC, even the patient’s level of consciousness. These real-time assessments enable the rapid detection of hemodynamic instability and the immediate identification of cardiac arrest, facilitating the early initiation of CPR and the optimization of resuscitation efforts [19]. The presence of skilled personnel and access to advanced equipment, such as defibrillators and emergency medications, further ensures the delivery of high-quality, timely interventions [18,19,20]. By integrating advanced monitoring with rapid access to resources, the OR environment minimizes delays and enhances the precision and effectiveness of resuscitative efforts, contributing to reduced CPR duration and improved survival outcomes [44,45].

Hypovolemia due to massive blood loss was identified as the leading cause of POCA in our study, accounting for 74.30% of cases, consistent with prior research [43] that highlights its critical role in surgical emergencies. The high prevalence of hemorrhagic shock emphasizes the necessity of continuous hemodynamic and respiratory monitoring for the early detection of instability. The real-time tracking of parameters such as blood pressure, oxygen saturation, ETCO_2_, and HR facilitates the identification of reversible conditions, such as hypovolemia and hypoxia, allowing for timely interventions, including fluid resuscitation, blood transfusion, and ventilatory adjustments, to prevent progression to cardiac arrest [12,43]. The OR offers a uniquely equipped environment to manage such crises, with advanced monitoring systems, healthcare teams trained in advanced cardiovascular life support (ACLS), and basic life support (BLS), and immediate access to defibrillators, blood products, and medications. These resources, combined with vigilant surveillance, enable clinicians to rapidly address the root causes of hemodynamic instability, particularly in trauma and high-risk surgeries, where significant blood loss is anticipated. The integration of early detection, continuous monitoring, and targeted interventions underscores the importance of a proactive approach to perioperative management, ultimately enhancing survival outcomes in high-risk scenarios [3,12,13,18,19,20,22,23].

This study has several limitations. First, as a retrospective, single-center analysis, the findings may not be generalizable to institutions with varying resources, protocols, or patient populations. Second, the absence of an electronic medical record (EMR) system during the study period posed challenges in ensuring consistent and precise data collection, particularly for continuous monitoring parameters during POCA events. This reliance on manually recorded data may have introduced variability, especially in documenting vital signs and other critical parameters in real time. Additionally, predictors such as preoperative SPO_2_ and ETCO_2_ were measured upon arrival to the OR and may not accurately represent the patient’s physiological state during the POCA event itself, as these parameters can fluctuate due to intraoperative management. Another limitation is that this study focuses solely on 24 h survival outcomes, leaving questions about long-term survival, the recurrence of POCA, and post-resuscitation complications unanswered. Future research should prioritize multi-center studies.

## 5. Conclusions

Despite the low 24 h survival rate following POCA, this study identifies key predictors that can inform clinical decision making and improve outcomes. These predictors include ASA 1–2, preoperative hemoglobin ≥ 8 g/dL, SpO_2_ ≥ 90%, initial ETCO_2_ levels between 35 and 45 mmHg, CPR duration ≤ 30 min, no preoperative sign of shock, and non-trauma surgeries. Hypovolemia due to hemorrhagic shock was the leading cause of mortality, emphasizing the need for early detection and appropriate fluid and blood replacement. The findings from this study should be further developed into predictive models and validated in multi-center, prospective studies to ensure their generalizability across diverse clinical settings. Such efforts are crucial for optimizing perioperative cardiac arrest management and improving both survival rates and long-term patient outcomes.

## Figures and Tables

**Figure 1 jcm-14-00599-f001:**
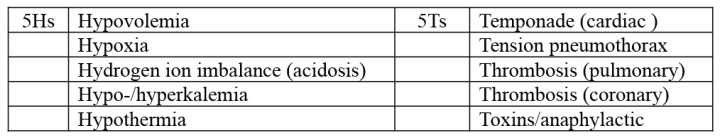
Cause of cardiac arrest.

**Figure 2 jcm-14-00599-f002:**
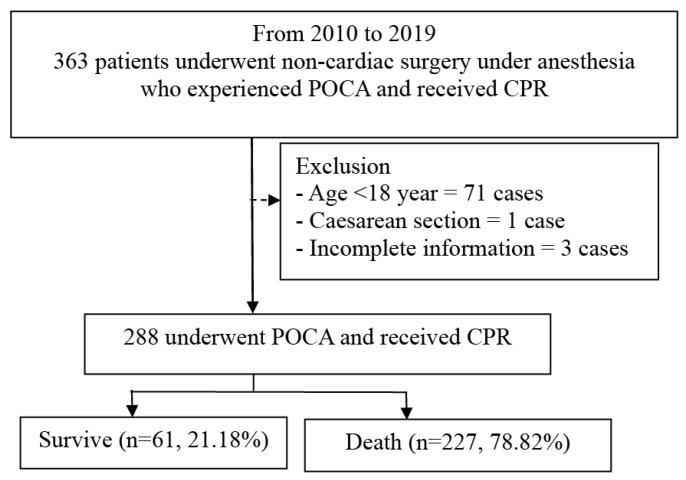
Study flowchart. Abbreviations: POCA (perioperative cardiac arrest); CPR (cardiopulmonary resuscitation).

**Figure 3 jcm-14-00599-f003:**
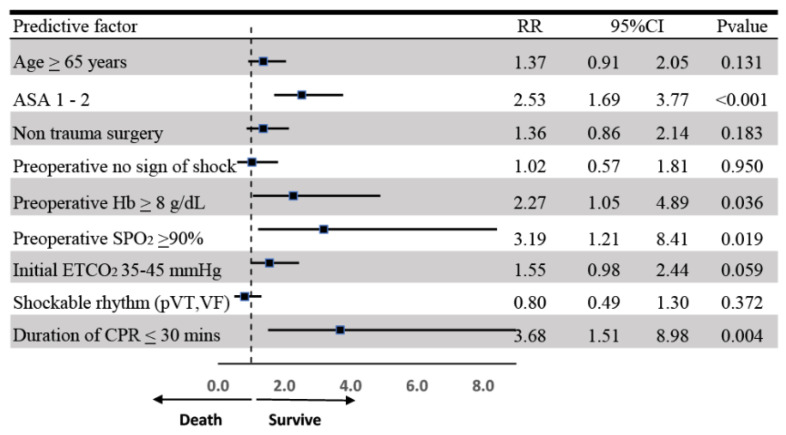
Multivariable risk regression analysis pertinent to survival within 24 h after perioperative cardiopulmonary resuscitation. Abbreviations: ASA: American Society of Anesthesiologists, Hb: hemoglobin, SPO_2_: oxygen saturation, ETCO_2_: the end-tidal carbon dioxide that was measured immediately following successful intubation, pVT: pulseless ventricular tachycardia, VF: ventricular fibrillation, CPR: cardiopulmonary resuscitation.

**Table 1 jcm-14-00599-t001:** Baseline characteristics of the study population and univariable analysis of prognostic factors for 24 h survival after perioperative CPR (*n* = 288).

Variables.	Survival (*n* = 61)	Death (*n* = 227)	RR	95%CI	*p*-Value
**Age (years)**					
** ≥65, *n* (%)**	28 (45.90)	69 (30.40)	1.67	1.07–2.59	0.024 *
** <65, *n* (%)**	33 (54.10)	158 (69.60)	Ref.		
**Mean ± SD**	59.16 ± 19.77	53.09 ± 20.03			
**Gender**					
** Female, *n* (%)**	27 (44.26)	67 (29.52)	1.63	1.05–2.55	0.028 *
** Male, *n* (%)**	34 (55.74)	160 (70.48)	Ref.		
**ASA physical status, *n* (%)**					
** 1–2**	20 (32.79)	6 (2.64)	4.92	3.46–6.99	<0.001 *
** 3–5**	41 (67.21)	221 (97.36)	Ref.		
**Non-smoker, *n* (%)**					
** Yes**	52 (85.25)	189 (83.26)	1.13	0.59–2.13	0.713
** No**	9 (14.75)	38 (16.74)	Ref.		
**Preoperative comorbidities, *n* (%)**					
** Diabetes mellitus**	9 (14.75)	28 (12.33)	1.17	0.63–2.18	0.611
** Hypertension**	29 (47.54)	57 (25.11)	2.13	1.38–3.29	<0.001 *
** Congestive heart failure**	3 (4.92)	14 (6.17)	0.82	0.28–2.36	0.720
** Ischemic heart disease**	6 (9.84)	15 (6.61)	1.39	0.67–2.84	0.371
** Respiratory disease**	9 (14.75)	28 (12.33)	1.17	0.63–2.18	0.611
** Renal disease**	15 (24.59)	29 (12.78)	1.81	1.11–2.94	0.017 *
** TIA/stroke**	5 (8.20)	13 (5.73)	1.34	0.61–2.93	0.464
** Sepsis**	4 (6.56)	13 (5.73)	1.12	0.46–2.72	0.805
** Solid cancer**	13 (21.31)	14 (6.17)	2.62	1.64–4.18	<0.001 *
**Current medication, *n* (%)**					
** Anticoagulant**	15 (24.59)	28 (12.33)	1.86	1.14–3.01	0.012 *
** Antihypertensive drug**	20 (32.79)	55 (24.23)	1.38	0.87–2.21	0.170
** Steroid**	7 (11.48)	9 (3.96)	2.20	1.20–4.04	0.011 *
** Insulin**	2 (3.28)	3 (1.32)	1.92	0.64–5.76	0.245
**Preoperative sign of shock, *n* (%)**					
** No**	49 (80.33)	132(58.15)	2.41	1.34–4.33	0.003 *
** Yes**	12 (19.67)	95 (41.85)	Ref.		
**Preoperative mechanical ventilation, *n* (%)**					
** No**	22 (36.07)	48 (21.15)	1.76	1.12–2.75	0.014 *
** Yes**	39 (63.93)	179 (78.85)	Ref.		

Values are presented as median (Q1, Q3), number (%), or mean ± SD. Abbreviations: ASA: American Society of Anesthesiologists. *: *p*-value ≤ 0.05 shows statistical significance.

**Table 2 jcm-14-00599-t002:** Details of surgical procedures and univariable analysis of prognostic factors for 24 h survival following perioperative CPR (*n* = 288).

Variables	Survival (*n* = 61)	Death (*n* = 227)	RR	95%CI	*p*-Value
Elective surgery, *n* (%)					
Yes	16 (26.23)	14 (6.17)	3.06	1.99–4.69	<0.001 *
No	45 (73.77)	213 (93.83)	Ref.		
Site of operation, *n* (%)					
Upper intra-abdominal	16 (26.23)	83 (36.56)	0.67	0.40–1.14	0.132
Major vascular	5 (8.20)	42 (18.50)	0.45	0.19–1.08	0.053
Intracranial	9 (14.75)	37 (16.30)	0.91	0.48–1.72	0.769
Intrathoracic	11 (18.03)	38 (16.74)	1.07	0.60–1.91	0.811
Orthopedic	7 (11.48)	9 (3.96)	2.20	1.20–4.03	0.023 *
Others	13 (21.31)	20 (8.81)	2.09	1.27–3.43	0.006 *
Non-trauma surgery, *n* (%)					
Yes	40 (63.57)	99 (43.61)	2.04	1.26–3.28	0.003 *
No	21 (34.43)	128 (56.39)	Ref.		
Supine position, *n* (%)					
Yes	49 (83.05)	212 (93.39)	2.13	1.24–3.67	0.006 *
No	10 (16.95)	15 (6.61)	Ref.		
Intraoperative blood loss, *n* (%)					
≤3000 mL	50 (81.97)	166 (73.13)	1.52	0.84–2.75	0.172
>3000 mL	11 (18.03)	61 (26.87)	Ref.		
Median [Q1, Q3]	500 [200, 2000]	1500 [200, 3900]			

Values are presented as median (Q1, Q3), number (%), or mean ± SD, RR: risk ratio, CI: confidence interval, SD: standard deviation. *: *p*-value ≤ 0.05 shows statistical significance.

**Table 3 jcm-14-00599-t003:** Preoperative laboratory investigation and univariable analysis of prognostic factors for 24 h survival after perioperative CPR (*n* = 288).

Variables	Survival (*n* = 61)	Death (*n* = 227)	RR	95%CI	*p*-Value	Missing Data, *n* (%)
Blood glucose level mg/dL, *n* (%)					35 (12.51)	
≤70	1 (2.08)	14 (6.90)	1.97	0.19- 20.26	0.570	
71–239	45 (93.75)	132 (65.02)	7.50	1.88–29.97	0.004 *	
≥240	2 (4.17)	57 (28.08)	Ref.			
Mean ± SD	147.23 ± 66.59	192.41 ± 98.19				
**Hemoglobin ≥ 8 g/dL, *n* (%)**						3 (1.04)
Yes	53 (86.89)	130 (58.04)	3.69	1.83–7.47	<0.001 *	
No	8 (13.11)	94 (41.96)	Ref.			
Mean + SD	10.59 ± 2.59	8.83 ± 3.58				
Platelet count ≥ 100 × 10^3^ cells/µL, *n* (%)						3 (1.04)
Yes	54 (88.52)	151 (67.41)	3.01	1.43–6.34	0.004 *	
No	7 (11.48)	73 (32.59)	Ref.			
Median [Q1, Q3]	201 [153, 291]	137 [86.5, 217]				
Neutrophil–to-lymphocyte ratio (NLR) ≤ 9, *n* (%)						3 (1.04)
Yes	42 (70)	166 (74.11)	0.85	0.52–1.39	0.520	
No	18 (30)	58 (25.89)	Ref.			
Median [Q1, Q3]	5.63 [2.62, 12.38]	4.71 [2.37, 10.13]				
**Serum albumin > 3.5 g/dL, *n* (%)**					22 (7.66)	
Yes	21 (37.50)	49 (23.33)	1.68	1.05–2.68	0.030 *	
No	35 (62.50)	161 (76.67)	Ref.			
Mean ± SD	3.06 ± 0.92	2.46 ± 1.13				
Serum creatinine < 2 mg/dL, *n* (%)						2 (0.69)
Yes	47 (77.05)	173 (76.89)	1.00	0.52–1.98	0.979	
No	14 (22.95)	52 (23.11)	Ref.			
Median [Q1, Q3]	1.2 [0.8, 1.9]	1.3 [0.9, 1.9]				
Sodium (mmol/L), *n* (%)						2 (0.69)
<135	8 (13.11)	26 (11.56)	3.59	1.21–10.42	0.026 *	
135–145	49 (80.33)	142 (63.11)	3.91	1.16–11.07	0.006 *	
>145	4 (6.56)	57 (25.33)	Ref.			
Mean ± SD	139.11 ± 5.38	141.84 ± 7.50				
Potassium (mmol/L), *n* (%)						2 (0.69)
≤3	9 (14.75)	41 (18.22)	2.07	0.68–6.28	0.199	
3.1–4.9	48 (78.69)	142 (63.11)	2.91	1.10–7.66	0.031 *	
≥5	4 (6.56)	42 (18.67)	Ref.			
Mean ± SD	3.82 ± 0 74	4.09 ± 1.19				

Values are presented as median (Q1, Q3), number (%), or mean ± SD, cells/µL: cells per microliter, g/dL: grams per deciliter, mg/dL: milligrams per deciliter. *: *p*-value ≤ 0.05 shows statistical significance.

**Table 4 jcm-14-00599-t004:** Initial vital signs upon arrival to the operating room and univariable analysis of prognostic factors for 24 h survival after perioperative CPR (*n* = 288).

Variables	Survival (*n* = 61)	Death (*n* = 227)	RR	95%CI	*p*-Value	Missing Data, *n* (%)
Heart rate (bpm), *n* (%)						8 (2.78)
>100	19 (31.15)	115 (52.51)	1.70	8.40–3.43	<0.001 *	
51–100	42 (68.85)	92 (42.01)	2.21	1.36–3.59	0.001 *	
≤50	0 (0)	12 (5.48)	Ref.			
Mean ± SD	96.62 ± 21.16	101.82 ± 28.58				
Systolic blood pressure (mmHg)						12 (4.16)
>140	13 (21.31)	35 (16.28)	3.72	1.47–7.47	0.004 *	
81–140	40 (65.57)	90 (41.86)	3.77	1.85–7.69	<0.001 *	
≤80	8 (13.11)	90 (41.86)	Ref.			
Mean ± SD	114.98 ± 26.69	98.65 ± 38.17				
Diastolic blood pressure (mmHg)						12 (4.16)
>80	8 (13.11)	97 (45.12)	3.94	1.74–8.93	0.001 *	
51–80	41 (67.21)	90 (411.86)	4.11	2.01–8.30	<0.001 *	
<50	12 (19.67)	28 (13.02)	Ref.			
Mean ± SD	69.18 ± 16.74	57.81 ± 25.15				
Mean arterial pressure (mmHg)						12 (4.16)
>95	14 (22.95)	41 (19.07)	2.88	1.33–6.24	0.007 *	
65–95	38 (62.30)	81 (37.67)	3.62	1.84–7.13	<0.001 *	
<65	9 (14.75)	93 (43.26)	Ref.			
Mean ± SD	84.44 ± 19.09	71.14 ± 28.45				
Preoperative oxygen saturation						12 (4.16)
≥90%	56 (91.80)	137(63.72)	4.82	1.99–11.61	<0.001 *	
<90%	5 (8.20)	78 (36.28)	Ref.			
Median [Q1, Q3]	99 [96, 100]	97 [86, 100]				
The initial end-tidal carbon-dioxide (mmHg)						37 (12.89)
35–45	16 (29.09)	18 (10.06)	2.41	1.53–3.80	<0.001 *	
<35	39 (70.91)	161 (89.94)	Ref			
Mean ± SD	29.34 ± 7.97	23.91 ± 8.91				

Values are presented as median (Q1, Q3), number (%), or mean ± SD, mmHg: millimeters of mercury, bpm: beats per minute. *: *p*-value ≤ 0.05 shows statistical significance.

**Table 5 jcm-14-00599-t005:** Details of POCA events and perioperative CPR, and univariable analysis of prognostic factors for 24 h survival following perioperative CPR (*n* = 288).

Variables	Survival (*n* = 61)	Death (*n* = 227)	RR	95%CI	*p*-Value
Time of cardiac arrest					
Working hours	33 (54.10)	87 (38.33)	1.65	1.05–2.58	0.028 *
Non-working hours	28 (45.90)	140 (61.67)	Ref.		
Initial rhythm at time of CPR					
Shockable rhythm (pVT/VF)	14 (22.95)	30 (13.22)	1.65	0.99–2.73	0.050 *
Non-shockable rhythm					
(PEA/asystole)	47 (77.05)	197 (86.78)	Ref.		
Place where the cardiac arrest occurred					
In the operating room	58 (95.08)	194 (85.46)	2.76	0.91–8.37	0.072
ICU/PACU	3 (4.92)	33 (14.54)	Ref.		
Duration of CPR					
≤30 min	57 (93.44)	160 (70.48)	4.66	1.75–12.42	0.002 *
>30 min	4 (6.56)	67 (29.52)	Ref.		
Median [Q1, Q3]	15 [5, 20]	20 [10, 35]			

Values are presented as median (Q1, Q3), number (%), or mean ± SD, RR: risk ratio, CI: confidence interval, SD: standard deviation, pVT: pulseless ventricular tachycardia, VF: ventricular fibrillation, ICU: intensive care unit, PACU: postanesthetic care unit. *: *p*-value ≤ 0.05 shows statistical significance.

**Table 6 jcm-14-00599-t006:** Cause of POCA (*n* = 288).

Cause of POCA	Total (*n* = 288)	Survive (*n* = 61)	Death (*n* = 227)
Hypovolemia	214 (74.30%)	37 (60.66)	177 (77.97)
Hypoxia	28 (9.72%)	9 (14.75)	19 (8.37)
Hydrogen ions (acidosis)	18 (6.25%)	5 (8.20)	13 (5.73)
Hypothermia	3 (1.04%)	0 (0)	3 (1.32)
Cardiac tamponade	1 (0.35%)	1 (1.64)	0 (0)
Tension pneumothorax	4 (1.39%)	1 (1.64)	3 (1.32)
Pulmonary thrombosis	1 (0.35%)	0 (0)	1 (0.44)
Coronary thrombosis	16 (5.56%)	5 (8.20)	11 (4.85)
Toxins/anaphylactic	3 (1.04%)	3 (4.92)	0 (0)

## Data Availability

Datasets generated or analyzed in this study are accessible upon reasonable request from the corresponding author.

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
