# Peer review of "Predictive Factors for 24-h Survival After Perioperative Cardiopulmonary Resuscitation: Single-Center Retrospective Cohort Study"

_jcm, 2025, doi:10.3390/jcm14020599_

Round 1
Reviewer 1 Report
Comments and Suggestions for Authors
The study addresses perioperative cardiac arrest (POCA), focusing on identifying predictors of 24-hour survival, which is clinically significant and directly relevant to improving patient outcomes. Despite the inherent limitations of its retrospective design, the study employs clear inclusion and exclusion criteria along with robust statistical analysis. Below are my suggestions.
1) Page 7, last paragraph - correct "greater than 90%" to "less than 90%"
2) The first line of the discussion on page 10 is hard to follow because of incorrect grammar and sentence structure. Please revise it.
3) The first line in the second paragraph of the discussion on page 10 is incomplete. Please revise.
The overall quality is acceptable, except for the corrections mentioned above to improve the flow.
Author Response
Comment 1: Page 7, last paragraph - correct "greater than 90%" to "less than 90%"
Response 1: Thank you for pointing this out. We agree with this comment. [Patients with SPO2 > 90% had a significantly higher survival rate compared to those with preoperative SPO2 measured by pulse oximetry while breathing room air less than 90%]
Comment 2: The first line of the discussion on page 10 is hard to follow because of incorrect grammar and sentence structure. Please revise it.
Response 2: Thank you for pointing this out. We agree with this comment. [This study found a 24-hour survival rate of 21% following perioperative CPR (Figure 1), emphasizing the critical nature of the condition. This survival rate is significantly lower than previously reported rates, which range from 32% to 51.4% in the OR settings within 24 hours after perioperative CPR [6, 10, 33, 34].]
Comment 3: The first line in the second paragraph of the discussion on page 10 is incomplete. Please revise.
Response 3: Thank you for pointing this out. We agree with this comment. [Previous studies indicate that survival rates for cardiac arrest are significantly higher in the OR compared to general wards or ICUs. This is primarily due to the swift recognition of patient condition changes through continuous monitoring and the prompt initiation of resuscitation measures [12, 35, 36]. ]
Thank You so much for your recommendation and kindness
Reviewer 2 Report
Comments and Suggestions for Authors
Dear authors,
I must congratulate all of you for compiling and analyzing the data and producing a good manuscript.
I am happy to review this submission.
I have requested a few technical edits to this submission.
Please address the comments embedded in the PDF as comments.
Best wishes.

Author Response
Comment 1: Page 3 , provide expansion "ASA"
Response 2: Thank you for pointing this out. We agree with this comment. [The American Society of Anesthesiologists physical status]
Comment 2: Page 3, Can you also summarize the 5H and T in a table with appropriate reference
Response 2: Thank you for pointing this out. We agree with this comment. and show with the [Figure 1. show mnemonic "5Hs and 5Ts," in a table]
Comment 3: Page 4, please mention the software/program used for sample size estimation
Response 3: Thank you for pointing this out. We agree with this comment. [The sample size was estimated using STATA version 16.0 (Stata Corp LP, College Station, TX, USA). A two-proportion test with an alpha level of 0.05 and a power of 0.80 indicated a requirement of 77 participants per group, resulting in a minimum total of 154 patients. This study included 288 eligible cases, ensuring robust statistical power to detect differences across predictor variables.]
Thank you so much for your recommendation and kindness
Reviewer 3 Report
Comments and Suggestions for Authors
This is a single center retrospective cohort study of the factors associated with post-surgical mortality after post operative cardiac arrest. The manuscript is well written and referenced. The findings are important because they provide insights into improved risk stratification prior to surgery. I have only minor comments tge should be addressed.
The title should include that it was a single-center retrospective cohort study.
Please alphabetize the keywords and make sure that they are found in MeSH.
Only the key r rings were summarized in the text from the tables. Thanks!
Limitations are well explained.
References are in MDPI style.
Congratulations on a fine contribution to the surgical emergencies literature.
Author Response
Comment 1: The title should include that it was a single-center retrospective cohort study.
Response 1: Thank you for pointing this out. We agree with this comment. [Predictive Factors for 24-Hour Survival After Perioperative Cardiopulmonary Resuscitation: single-center retrospective cohort study]
Comment 2: Please alphabetize the keywords and make sure that they are found in MeSH.
Response 2: Thank you for pointing this out. We agree with this comment. [Keywords: Cardiopulmonary resuscitation; Perioperative; cardiac arrest; Prognosis, Retrospective Studies; Survival]
Comment 3: Only the key r rings were summarized in the text from the tables
Response 3: Thank you for pointing this out. But are not sure what this sentence means ; The key findings were summarized in the text from the table, So I have restructured the sentence to be as follows: Page 5 ; [All patients who experienced with POCA had received general anesthesia, with only one initially receiving an infraclavicular brachial plexus block and two others undergoing MAC. Patients aged 65 years and older demonstrated a higher survival rate compared to those younger than 65 years (RR 1.67; 95%CI 1.07–2.59; p = 0.023). Patients with ASA physical status classification 1-2 had a higher likelihood of survival compared to those with ASA 3-5 (RR 4.92; 95%CI 3.46–6.99; p <0.001). Patients undergoing non-trauma surgery demonstrated a higher survival rate compared to trauma with multiple injuries surgery (RR 2.04; 95%CI 1.26–3.28; p = 0.003). Patients with no preoperative signs of shock before experiencing POCA had better survival rates than those who were in shock upon arrival in the operating room (RR 2.41; 95% CI 1.34–4.33; p = 0.003)] We have cut out sentences that are not particularly important, namely: [ The average age of patients experiencing POCA was 54.38 ± 20.09 years.] and [Patients undergoing elective surgery showed a higher survival rate compared to emergency surgery (RR 3.06; 95%CI 1.99–4.69; p <0.001),]
Thank You verymuch for your recommendation and kindness
Reviewer 4 Report
Comments and Suggestions for Authors
The subject of perioperative cardiopulmonary arrest is a topic of interest, especially in areas where the incidence is higher, and survival rates are lower compared to those reported in developed healthcare systems. Exploring this critical aspect of medicine over a 10-year period is commendable.
The introduction is well-structured; however, the sentence „Cardiac arrest itself is
defined as an absence of cardiac rhythm requiring cardiopulmonary resuscitation (CPR)
whether by closed chest compression or, in rare cases, open cardiac massage [6, 8-11].” needs revision. Cardiac arrest refers to the absence of circulation, not the absence of cardiac rhythm. As is well known, there is electrical activity in shockable rhythms and in PEA.
Additionally, the expression „ closed chest compression” is unusual in a rigorous scientific context. I recommend removing the word "closed" to align with the terminology used in current resuscitation guidelines, which should be referenced in the bibliography. Please ensure proper punctuation, and every sentence should begin with a capital letter (e.g., after ref 12).
On page 10, sentence „The patients aged 65 and older, particularly those undergoing elective or non-emergent surgeries. ” lacks a predicate. Please revise it. Similarly, the conclusions chapter should adhere to the principles and begin with a capital letter.
Overall, the study is comprehensive, the statistical analysis aligns with the working hypothesis, and the data is clearly presented. Although none of the favorable prognostic factors for survival are surprising, the conclusions are relevant.
Perhaps, in a future study, the authors could also consider identifying human error as a variable in relation to the occurrence of cardiac arrest.
For the current research, I recommend a minor revision, specifically addressing editing issues rather than the core content of the manuscript.
Author Response
Comment 1: Cardiac arrest itself is defined as an absence of cardiac rhythm requiring cardiopulmonary resuscitation (CPR) whether by closed chest compression or, in rare cases, open cardiac massage [6, 8-11].” needs revision. Cardiac arrest refers to the absence of circulation, not the absence of cardiac rhythm. As is well known, there is electrical activity in shockable rhythms and in PEA.
Response 1: Thank you for pointing this out. We agree with this comment. [Cardiac arrest refers to the absence of circulation, requiring cardiopulmonary resuscitation (CPR) whether by chest compression or, in rare cases, open cardiac massage [6, 8-11].]
Comment 2 : Additionally, the expression „ closed chest compression” is unusual in a rigorous scientific context. I recommend removing the word "closed" to align with the terminology used in current resuscitation guidelines,
Response 2: Thank you for pointing this out. We agree with this comment. [whether by chest compression or, in rare cases, open cardiac massage [6, 8-11].]
Comment 3 : which should be referenced in the bibliography. Please ensure proper punctuation, and every sentence should begin with a capital letter; developed countries
Response 3: Thank you for pointing this out. We agree with this comment. [Developed countries]
Comment 4: On page 10, sentence „The patients aged 65 and older, particularly those undergoing elective or non-emergent surgeries. ” lacks a predicate.
Response 4: Thank you for pointing this out. We agree with this comment. [Our study demonstrates that patients aged 65 and older, particularly those undergoing elective or non-emergent surgeries, have significantly higher survival rates. ]
Comment 5: the conclusions chapter should adhere to the principles and begin with a capital letter.
Response 5: Thank you for pointing this out. We agree with this comment. [Conclusions]
Thank You very much for your recommendation and I will continue my research.